

# Mat thickness associated with *Didymosphenia geminata* and *Cymbella* spp. in the southern rivers of Chile

Daniel Zamorano[1], Matías Peredo-Parada[2], Diana J. Lillo[3], Jorge Parodi[4] and Carolina A. Díaz[5]

[1] Centro de Investigación e Innovación para el Cambio Climático (CiiCC), Universidad Santo Tomás, Santiago, Chile
[2] Plataforma de Investigación en Ecohidrología y Ecohidráulica (Ecohyd), Santiago, Chile
[3] Agencia de Comunicación Científica Héurēka, Santiago, Chile
[4] Laboratorio de Biología Celular y Molecular aplicada, Vicerrectoría de Investigación y Postgrado, Universidad Mayor, Temuco, Chile
[5] Amakaik Consultoría Ambiental, Santiago, Chile

## ABSTRACT

*Didymosphenia geminata* is a diatom that can alter aquatic systems. Several investigations have shown as chemical, and hydraulic factors have a great influence on the proliferation of *D. geminata*, but the study of other microalgae that could be associated with it has been poorly addressed. The objective of this study is to evaluate the relationship between mat thickness, *D. geminata* and another taxon that produces mucilage, *Cymbella*, while also considering physical and chemical factors. For this, two samples were taken, one in the spring of 2013 and the other in the autumn of 2014, from eight rivers in central-southern Chile-South America, where the benthic community was characterized, and the thickness of the mat was measured. The results show that the mat thickness on sites with the presence of both taxa is doubled, and while sites with *D. geminata* presence showed mat peak on autumn, sites with *Cymbella* spp. presence showed on spring. Also, higher values of mat thickness associated with low cell densities of *D. geminata* and intermediate cell densities of *Cymbella* spp. Finally, physicochemical variables that better explain mat thickness are phosphorus and water temperature. An alternation process of mucilage production may explain these results by these taxa strongly related to physicochemical variables. The present study contributes evidence about the relationship between mat thickness *D. geminata* and other microalgae contribution, and aquatic condition for this development.

# INTRODUCTION

Since the beginning of the 2000s, massive mat proliferation in oligotrophic rivers by the extracellular stalk material of *Didymosphenia geminata* (Lyngbye) A. Schmidt has aroused the interest of scientists and governmental entities due to its economic impacts (*Beville, Kerr & Hughey, 2012*) and its ability to severely affect conditions in aquatic systems (*Spaulding & Elwell, 2007*). For example, *D. geminata* mats can alter river community composition and food web structure, favoring some species over others

Corresponding author
Matías Peredo-Parada,
matias.peredo@ecohyd.com

(*Kilroy, Larned & Biggs, 2009*; *Gillis & Chalifour, 2010*; *Ladrera, Gomà & Prat, 2018*). Additionally, in periphytic algae, the mat matrix allows carbon fixation at rates typical of higher nutrient inputs (*Aboal et al., 2012*; *Tyler, McGlathery & Anderson, 2003*), and there is a link between stalk biomass and rates of phosphoester hydrolysis (*Aboal et al., 2012*; *Bray, O'Brien & Harding, 2017*). Likewise, *D. geminata* mats reduce stresses and turbulent velocity fluctuations on riverbeds, which may reduce the risk of mat detachment (*Larned et al., 2011*), persisting up to one month following a peak in growth (*Miller et al., 2009*).

The massive growth of *D. geminata* mats has been related to chemical, physical and biological variables that control its presence or the conditions of its proliferation. For example, the presence of nitrogen-fixing cyanobacteria (*Novis, Schallenberg & Smissen, 2016*) and levels of dissolved reactive phosphorus (*Bothwell, Taylor & Kilroy, 2014*; *Bray et al., 2017*) favor *D. geminata* blooms, while an increase in riverbed shear stress favors mat removal (*Cullis, Crimaldi & McKnight, 2013*) as well as sediment transport, both of which are conditioned by increased flow (*Miller et al., 2009*).

Although other diatoms can generate stalks or tubes of mucilage (*Bahls, 2007*), the possible role of these diatoms as massive mat producers in a possible synergistic relationship with *D. geminata* has rarely been studied. Researchers have recorded other microalgae species able to produce mats through macroscopic visual records on rivers and analysis at the microscopic scale of stalks (*Bahls, 2007*; *Suzawa, Seino & Mayama, 2011*; *Khan-Bureau et al., 2014*; *Khan-Bureau et al., 2016*; *Furey, Kupferberg & Lind, 2014*). However, there are few publications to date that quantify the relationship between the microalgae community, microalgae density, and mat thickness.

Previous works have already identified species of the genus *Cymbella* as potentially harmful algae. In the United States, sites with gray mucilage associated with the species *C. janischii* (A. Schmidt) De Toni have been reported (*Bahls, 2007*; *Khan-Bureau et al., 2014*), and in Japan, a country that has not been invaded by *D. geminata*, algal blooms have also been found associated with *C. janischii* (*Suzawa, Seino & Mayama, 2011*). In several places in Chile, *Jaramillo et al. (2015)* recorded seven species of *Cymbella* together with the presence of *D. geminata* through molecular analysis without detecting the presence of *C. janischii*. The seven species were *C. affinis, C. cistula, C. mexicana, C. proximal, C. aspera, C. lanceolata* and *C. tumida* (Brébisson) van Heurck. In none of these studies was the mat thickness estimated to be related to the abundance of species of the genus *Cymbella*.

In light of this background, the objective of the present study is to evaluate the relationship of *D. geminata* and *Cymbella* spp. with mat thickness in rivers in the south-central zone of Chile. For this, three specific objectives are proposed: (I) quantify the mat thickness in different rivers, (II) relate mat thickness with the presence and density of *D. geminata* and *Cymbella* spp., and (III) evaluate the relationships among the physical and chemical conditions, mat thickness, *D. geminata* and *Cymbella* spp. presence and their density.

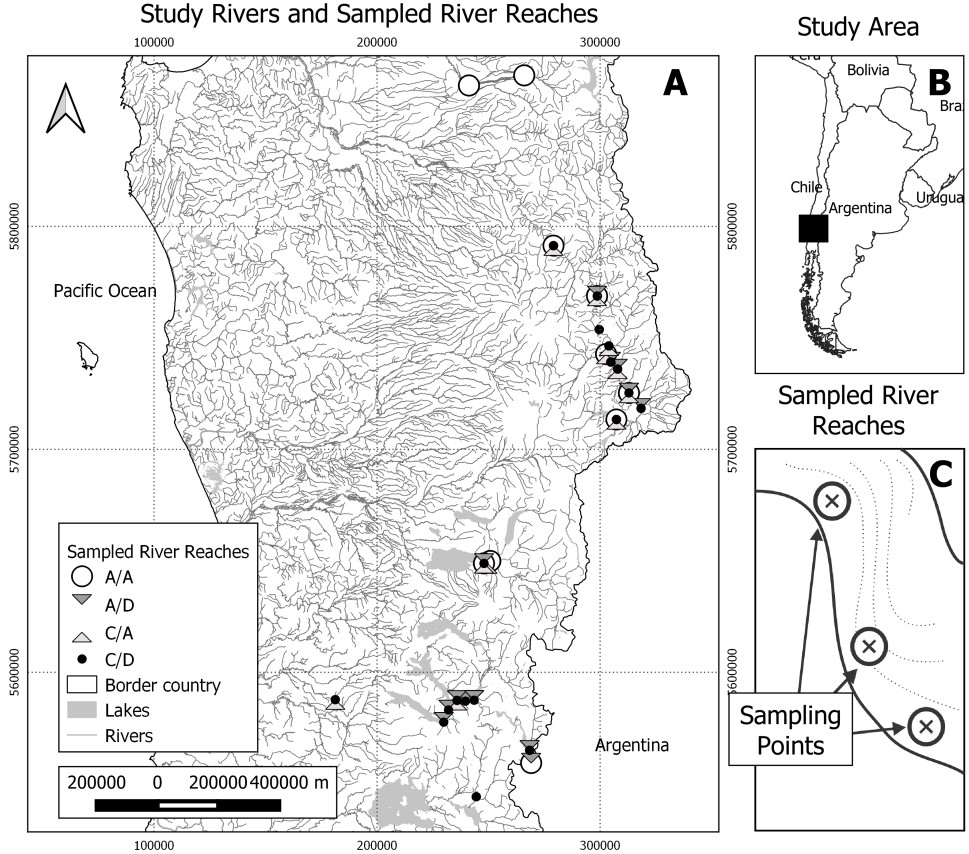

**Figure 1** **Study area and sampling scheme.** (A) Study rivers and sampled river reaches. A total of 25 river reaches were sampled during two seasons (spring and autumn). A/A, reaches with absence of both *D. geminata* and *Cymbella spp.*; A/D, reaches with the presence of *D. geminata* only; C/A, reaches with the presence of *Cymbella* spp. only; and C/D, reaches with the presence of both species. (B) Study area. The study was carried out in the regions of Bío-Bío, Araucanía and Los Ríos, in the south-central zone of Chile. Black rectangle represents map A area. (C) Representation of sampled river reaches. In each river reach between one and six sampling points were registered, with a total of 123 sampling points during both campaigns.

# MATERIALS AND METHODS

## Study area

The study was carried out in eight rivers located in four basins in the south-central zone of Chile (Fig. 1) with *D. geminata* presence records (*Montecino et al., 2016*). These rivers were selected because they were characterized by circumneutral pH values, cold and oligotrophic waters and substrates composed mainly of boulders, which are favorable conditions for *D. geminata* proliferation (*Kawecka & Sanecki, 2003*; *Spaulding & Elwell, 2007*; *Bray, O'Brien & Harding, 2017*). Additionally, these rivers were selected due to the possibility of access and absence of anthropogenic hydrological regulations (dams, large intakes, among others).

Within these rivers, 24 short river reaches of 50 m were identified and georeferenced. The minimal distance between the selected reaches was 2 km. In each river reach, we

selected between one and six sampling points that represented all the hydromorphological diversity of the river reach, prioritizing the sampling of all the mesohabitat types detected (*Alcaraz-Hernández et al., 2011*). We randomly selected one boulder for each sampling point to obtain the biological samples, which represents the unit of analysis of this study (Fig. 1).

The sampling was carried out on two occasions: late spring (sampling 19 river reaches, 74 sampling points, December 2013) and early autumn (sampling 17 river reaches, 49 sampling points, March 2014). Due to access problems and high flow, only 12 of the 24 river reaches were sampled in both seasons. Given that the hydraulic dynamics of these 12 rivers did not allow the sampling of the same point twice, these sampling points were considered statistically independent from each other. During the late spring season, the water temperature varied between 9 °C and 18 °C, with an average conductivity of 45.7 $\mu$S/cm. Meanwhile, the fall season had lower temperatures between 4 °C and 12 °C and a higher average conductivity (52.6 $\mu$S/cm).

## Sampling methods

Mat thickness was measured with a graduated ruler at each sampling point. To evaluate the microalgae community at each sampling point, one algal biofilm sample of 1 mL was taken from the mat using a disposable blunt syringe. If the mat layer had a thickness of less than 1 mm, an area of 4 $cm^2$ was swept with a disposable brush, assuming a volume of 0.4 mL. The samples were fixed with 1 mL of Lugol's iodine in water and stored in sealed 15 mL containers.

The biological samples were examined quantitatively in the laboratory. For this, a subsample of 1 mL was obtained and deposited in a Sedgewick-Rafter chamber and analyzed using a Zeiss Axiostar II Plus microscope with inverted objective microscopy set at 40×. We counted almost 100 valves for identification. However, in low abundances samples, we sampled almost three transects of 20 squares of 1 $mm^2$. The identification was performed up to the genus taxonomic level. We used the following works for the taxonomic identifications: (*Bourrelly, 1970*; *Parra et al., 1982*; *Cox, 1996*; *Round, Crawford & Mann, 1996*; *Biggs & Kilroy, 2000*; *Bicudo & Menezes, 2006*; *Sant'Anna et al., 2006*). Finally, richness was calculated with the total taxa detected at the sampling point.

To confirm the identification of *D. geminata*, the samples were oxidized with sulfuric acid based on the methodology described in *Battarbee (1986)*. Once the permanent preparations were obtained, the sample was examined using an Axisostar II Plus optical microscope at 1,000×.

Upstream of each river reach sampled, the following physical and chemical parameters were measured using a HANNA Model HI 9828 multiparameter probe: temperature, pH, electrical conductivity (EC) and dissolved oxygen (DO). In addition, samples were taken to estimate the levels of calcium (Ca), total phosphorus (P), iron (Fe) and silicate ($SiO_2$) through the analysis protocols of ''Standard Methods for Examination of Water & Wastewater'' (*Apha, 2005*). These samples were taken to the laboratory within 48 h. At each sampling point, the water column depth (depth) was measured using a graduated bar, and the superficial, middle and background flow velocities were measured using a Global

Water FP 101 digital flowmeter. For statistical analysis, physical-chemical values lower than the detection limit were transformed to a value between 0 and the detection limit for each variable according to the suggestion by *Helsel & Cohn (1988)*.

## Analysis of *D. geminata* and *Cymbella* spp. versus mat thickness

A two-way ANOVA with permutations was performed to determine differences in the mat thicknesses between points with and without *D. geminata* and *Cymbella* spp. Two-way ANOVAs were applied from the methodology used by *Anderson & Legendre (1999)*. Then, a posteriori nonparametric Tukey test was performed using the "mctp" program with the "narpcomp" package (*Konietschke et al., 2015*). The samples were categorized as presence only (C/A—*Cymbella* spp. alone or A/D—*D. geminata* alone), the presence of both (C/D) and the absence of both taxa (A/A).

A smooth surface model was fitted to evaluate the relationship between mat thickness (independent variable) and the cell densities of *D. geminata* and *Cymbella* spp. (predictor variables). The model was adjusted using the "oridsurf" function (*Marra & Wood, 2011*) from the "vegan" package (*Oksanen et al., 2015*) and selecting the REML method (restricted maximum likelihood) to estimate the model smoothing parameters.

Finally, to complement these analyses and consider the temporal scale, we made different plots to compare sampling seasons (spring 2013 and autumn 2014), cell densities and mat thickness according to the presence or absence of *D. geminata* and *Cymbella* spp.

## Analysis of *D. geminata* and *Cymbella* spp. versus the physical and chemical variables

To relate the physical and chemical conditions to a temporal scale, we carried out a MANOVA with permutation comparing these parameters between the sampling seasons. Then, ANOVA with permutation was performed for each physical and chemical variable, comparing between the sampling seasons. A MANOVA permutation test and posterior test were carried out using the "RVAideMemoire" package (*Hervé, 2015*). ANOVA was applied from the methodology used by *Anderson & Legendre (1999)*.

Second, through the one-way ANOVAs and MANOVA with permutations, the values of the physical and chemical parameters were compared according to the categories A/A, A/D, C/A and C/D. A pairwise nonparametric test was performed on the posterior significate. Additionally, linear discriminant analysis (LDA) (*Venables & Ripley, 2002*; *Ripley, 2007*) was performed to determine which of these parameters explained the multivariate context (MASS package; *Venables & Ripley, 2002*). Additionally, the physical and chemical parameters with significant differences were correlated with the total richness of the taxa of each sampling point. According to the temporal and categorical results, plots were generated to complement the analysis.

## Analysis of mat thickness versus the other variables

A random forest (RF) regression model was adjusted to determine the explanatory ability of the physical, chemical, temporal and biological variables on mat thickness, considering all the physical and chemical parameters, sampling seasons, presence or absence and density of *D. geminata* and *Cymbella* spp. The random forest analysis (*Breiman, 2001*) was

performed using the "caret" package (*Kuhn, 2008*). This model was trained using a 5-fold cross-validation scheme. The caret package allows for the fitting and tuning of models. Many predictive and machine learning models have structural or hyperparameters that cannot be directly estimated from the data. For RF models, classification trees may be built using a given number of randomly selected predictors that are named "*mtry*" (*Kuhn & Johnson, 2013*). A hyperparameter such as *mtry* is usually fixed at a given value when training and calibrating an RF model, which is itself an iterative optimization process. The hyperparameter tuning of an RF model refers to a grid search procedure that allows the algorithm to find the best value of *mtry* to obtain the best model performance (given a set of calibration and validation data points). In our implementation of RF models, the search for an optimal *mtry* value spanned the space between 2 and the total number of variables. Thus, the tuning process allowed us to explore a range of values for the RF hyperparameter, further improving the model performance. This generated a final model with the best hyperparameter value for a given search grid, which was selected according to the RMSE parameter (*Kuhn & Johnson, 2013*).

All the analyses were performed using R project software 3.3.0 v (*R Core Team, 2017*). In the supplemental material, a complete database (Table S1), *Cymbella* spp. and *D. geminata* microscopic photos and mat macroscopic photos (Fig. S1) can be found.

## RESULTS

### Mat thickness and taxa

The average mat thickness was 0.46 cm, with a maximum of 3 cm and a minimum of 0 cm. In 74% of the samples, a visible mat was detected (>0.1 mm); 78% of the sampling points had a mat between 0 cm and 1 cm, and only 10% of the sampling points had a mat greater than 1 cm.

Of the 123 points sampled, a detection of cells by microscopic counts occurred for *D. geminata* in 89 points (71%), 25 in autumn (20%) and 64 in spring (51%), and for *Cymbella* spp., 76 points were detected (60%), 28 in autumn (22%) and 48 in spring (38%). By category, the A/A sampling points occurred 15 times in autumn (12.2%) and 6 times in spring (4.8%), the C/A sampling points occurred 9 times in autumn (7.3%) and 4 times in spring (3.2%), the A/D sampling points occurred 6 times in autumn (4.8%) and 20 times in spring (16.3%), and finally, the C/D sampling points occurred 19 times in autumn (15.4%) and 44 times in spring (35.7%).

### Mat relationship with *D. geminata* and *Cymbella* spp.

Of the C/A sampling points, 75% did not exceed 3 mm in mat thickness, with a median of 0 mm, without visible mucilage. In contrast, the median thickness of the A/D sampling points was 3 mm, and the median thickness of the C/D sampling points was 5 mm. Thus, *Cymbella* spp., in the absence of *D. geminata*, recorded thinner mats than *D. geminata*, and in the absence of *Cymbella* spp., both species together recorded larger mat thicknesses (Fig. 2A).

The ANOVA with permutation showed significant differences between the categories ($p$ value = 0.02; $p$ perm = 0.02) (Table S1). The posterior Tukey test indicated significant
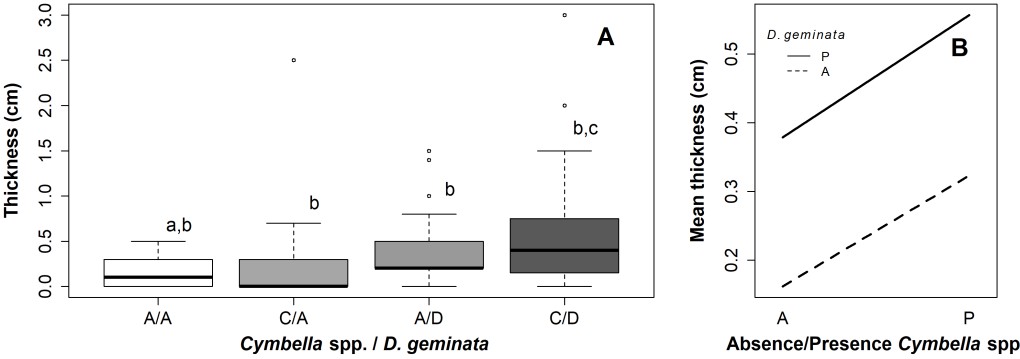

**Figure 2** **Graphs of the relationship between mat thickness and presence/absence of *Cymbella spp.* and *D. geminata*.** (A) Boxplot thickness in the presence (P) or absence (A) of *Cymbella sp.* and *D. geminata*. Letters above (a, b and c) represent the homogeneous groups of the Tukey test. (B) Graph of the interaction between presence and absence of *Cymbella spp.* and *D. geminata* against the average mat thickness.

differences between the A/A and C/D sampling points (*p* value = 0.01). Categories C/A and A/D did not differ significantly from the remaining points (A/A–C/A, *p* value = 0.96; A/A–A/D, *p* value = 0.16; C/A–A/D, *p* value = 0.6; A/D–C/D, *p* value = 0.64; and C/A–C/D, *p* value = 0.19) (Figs. 2A and 2B).

The relationship between the mat thickness and the cell density of *D. geminata* (Fig. 3B) and *Cymbella* spp. (Fig. 3A) showed an inverse pattern, being less clear in the *Cymbella* spp. case. Additionally, the relationship of densities between *Cymbella* spp. and *D. geminata* was inverse, and higher values of mat thickness were associated with low cell densities of *D. geminata* and intermediate cell densities of *Cymbella* spp. (Fig. 3C). Finally, the smooth surface of the mat thickness fitted to the relationship of *Cymbella* spp.- *D. geminata* explained 8.3% of the deviation, which is considered significant ($r^2 = 0.058$, *p* value = 0.035).

When comparing between the sampling seasons, the results suggested that the *Cymbella* spp. cellular density was always greater at the C/D sampling points, regardless of the season (Figs. 4A and 4D). In contrast, the *D. geminata* cellular density showed alternation over the A/D sampling points, being greater in spring and lower in autumn. In the C/D sampling points, the *D. geminata* density was always relatively high during both seasons (Figs. 4B and 4E). In spring, the A/D, C/A and C/D sampling points showed high mat thicknesses, and in autumn, only the A/D sampling points indicated this result (Figs. 4C and 4F).

## Participation of the physical and chemical variables over *D. geminata* and *Cymbella* spp.

MANOVA with permutations detected significant differences between the sampling seasons when considering the physical and chemical variables (*p* value = 0.001). The variables that significantly varied between the seasons according to the ANOVAs were P (mean: 0.008 mg/L in autumn; 0.13 mg/L in spring), temperature (mean: 8.4 °C in autumn; 13.7 °C in spring), EC (mean: 52.5 μS/cm in autumn; 45.7 μS/cm in spring), DO (mean: 9.2 mg/L in autumn; 10.3 mg/L in spring), Ca (mean: 6.8 mg/L in autumn; 6.0 mg/L in spring), Fe

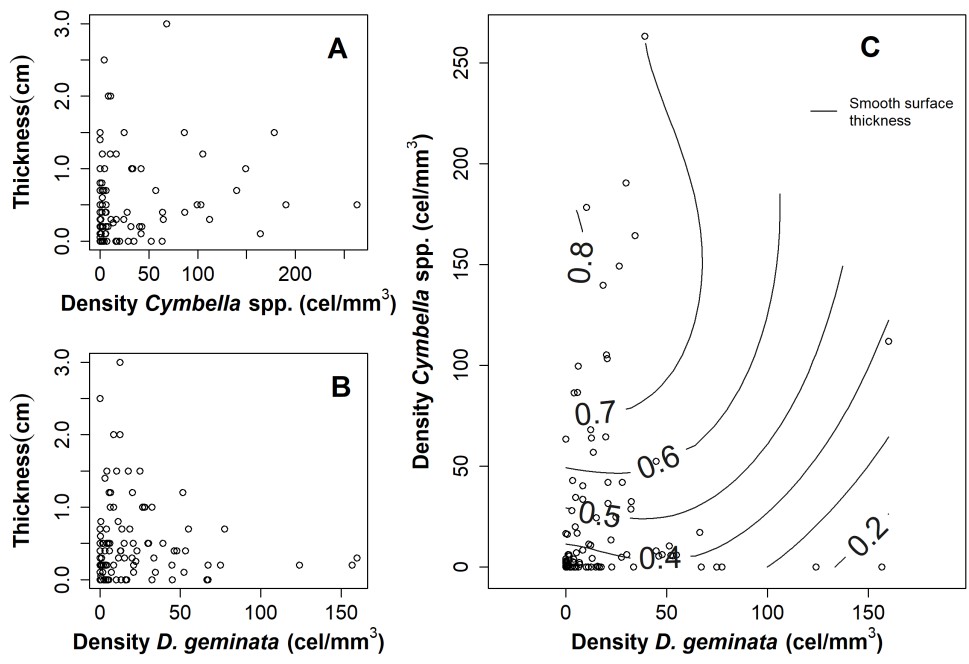

**Figure 3** **Graphs of the relationship between mat thickness and cell density of *Cymbella spp.* and *D. geminata*.** Mat thickness as a function of cell density of (A) *Cymbella* spp. and (B) *D. geminata*. (C) Smooth Surface model of mat thickness in cell density of *Cymbella spp.* versus cell density of *D. geminata* plot. Smooth Surface is significant ($p$ value = 0.03) and accounts for 8.3% of the variance.

(mean: 0.15 mg/L in autumn; 0.07 mg/L in spring) (results ANOVA with permutations in Table S1).

When evaluating the differences in the physical and chemical conditions between the A/A, C/A, A/D and C/D categories, MANOVA with permutations detected significant differences between the categories ($p$ value = 0.01). A posterior pairwise test indicated significant differences between the A/A and C/D and between the A/A and A/D sampling points (A/A–C/D, $p$ value = 0.003; A/A–A/D, $p$ value = 0.003). Category C/A did not differ significantly from the rest (A/A–C/A, $p$ value = 0.34; C/A–A/D, $p$ value = 0.43; and C/A–C/D, $p$ value = 0.11).

Regarding the LDA to discriminate between the A/A, C/A, A/D and C/D sampling points according the physical and chemical variables, axis 1 explained 85% of the variance. On axis 1, discrimination is maximized to the A/A and C/A sampling points versus the A/D and C/D sampling points (mean on axis 1: A/A = −2.06 , C/A = −0.59 , A/D = 0.17 and C/D = 0.60). When comparing these values with Table 1, P was strongly related to the A/A and C/A sampling points (the most negative variable), while pH and temperature were related to the A/D and C/D sampling points (the most positive variables) (Table 1) (Fig. 5).

When evaluating the differences between the categories by variable, the pH, DO, Ca, Fe, depth and background velocity did not change significantly. On the other hand, the variables that varied between the categories were temperature ($p$ ANOVA <0.001, $p$ perm = 0.002), EC ($p$ ANOVA < 0.001, $p$ perm = 0.001), P ($p$ ANOVA = 0.001, $p$ perm = 0.001)

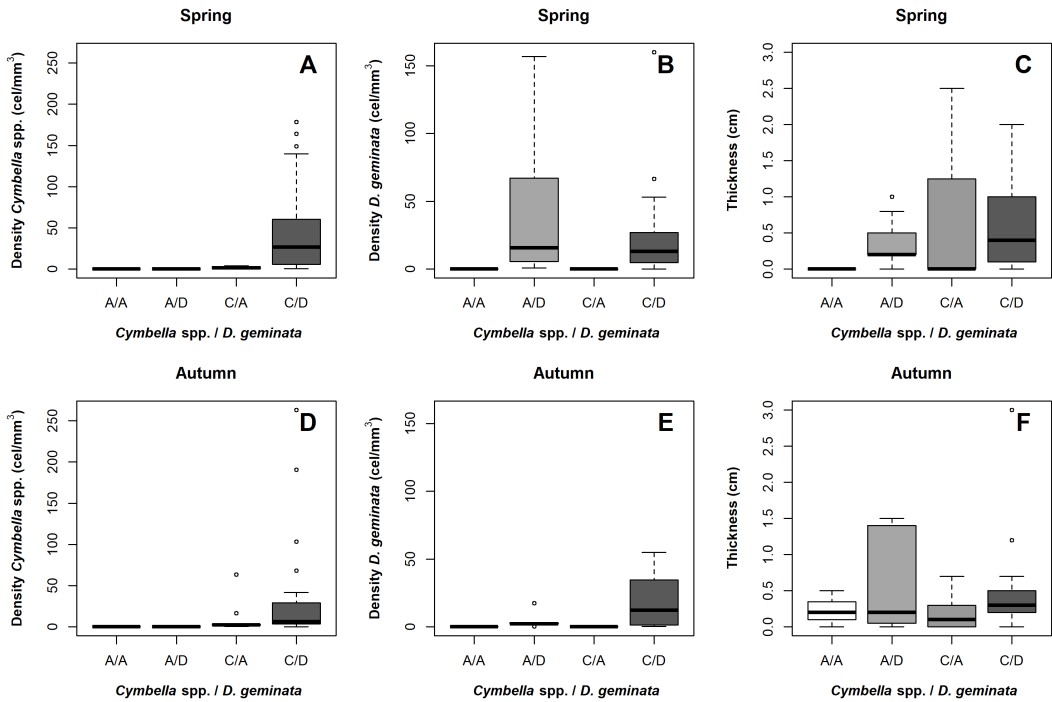

**Figure 4** **Boxplot of cellular density and mat thickness according to the presence/absence of *D. geminata* and *Cymbella spp* and sampling season.** Samples were categorized as presence only (C/A, *Cymbella* spp. alone) or (A/D, *D. geminata* alone); the presence of both (C/D) and, the absence of both taxa (A/A). (A) *Cymbella* spp. density on spring. Sites on presence of *D. geminata* and *Cymbella* spp. showed a peak of *Cymbella* spp. density. (B) *D. geminata* density on spring. Sites on presence of *D. geminata* were higher density. (C) Mat thickness on spring. Only A/A sites showed lower mat thickness. (D) *Cymbella* spp. density on autumn. Sites on presence of *D. geminata* and *Cymbella* spp. showed a peak of *Cymbella* spp. density. (E) *D. geminata* density on autumn. Sites on the presence of *D. geminata* and *Cymbella* spp. showed higher density. (C) Mat thickness on autumn. All sites showed similar thickness, being higher on A/D sites.

and $SiO_2$ ($p$ ANOVA $< 0.001$, $p$ perm $= 0.001$) (details of the ANOVAs and MANOVA in Table S1).

When performing the subsequent Tukey test, the water temperature varied between the categories, highlighting the differences between the A/A and the A/D–C/D categories (A/A–A/D, $p$ value $= 0.004$; and A/A–C/D, $p$ value $= 0.011$) and the differences between the C/A and the A/D–C/D categories (C/A–A/D, $p$ value $= 0.019$; and C/A–C/D, $p$ value $= 0.049$). At the temporal scale, while the A/A sampling had colder temperatures in spring, the autumn temperature was similar at all the sampling points (Fig. 6). For $SiO_2$, only differences between the A/A and the *D. geminata* presence sampling points were detected (A/A–A/D, $p$ value $<0.001$; and A/A–C/D, $p$ value $< 0.001$). On a temporal scale, the $SiO_2$ peak in the A/A sampling point during autumn is highlighted. For P and EC, the only differences occurred between the A/A and C/A sampling points (P: A/A–C/D, $p$ value $= 0.015$; and EC: A/A–C/D, $p$ value $= 0.054$). Similar to the silica case, the EC showed a peak at the A/A sampling points in autumn. For phosphorous, the sampling points showed

**Table 1 Results of Linear Discriminate Analysis (LDA) and Random Forest.** Column two shows participation by each predictor variable on the first axis of LDA analysis. Column three shows the participation percentage of each variable in the model generated by Random Forest. Column two, LDA to discriminate physical and chemical variables between presence/absence *D. geminata* and *Cymbella* spp. Categories (A/A–C/A–A/D–C/D); Phosphorus and temperature were the most relevant variables. Column three, Random Forest to determine the discriminatory capacity of physical, chemical and biological variables on mat thickness; most relevant variables were *D. geminata* density, *Cymbella* spp. Density, pH, water column depth and $SiO_2$.

| Predictor variables | LDA 1, between A/A–C/A–A/D–C/D | Random Forest, mat thickness response variable |
| --- | --- | --- |
| Ca | −0.54 | 14.61 |
| EC | −0.02 | 6.87 |
| Fe | 0.42 | 3.75 |
| DO | 0.37 | 11.90 |
| P | **−0.82** | **100.00** |
| pH | **0.50** | **28.44** |
| Water column depth | −0.29 | **23.63** |
| $SiO_2$ | −0.42 | **26.38** |
| Temperature | **0.53** | 12.58 |
| Background flow velocity | −0.10 | 7.43 |
| Sampling season | – | 0.86 |
| *Cymbella* spp. cell density | – | **29.32** |
| *D. geminata* cell density | – | **32.38** |
| *Cymbella* spp. Presence/absence | – | 8.23 |
| *D. geminata* Presence/absence | – | 7.21 |

higher values in spring and lower values in autumn, but in general, the A/A sampling points always showed higher values (Fig. 6).

The correlations between these four parameters and the total richness of the taxa showed that only $SiO_2$ was inversely and significantly related to the total richness, but not the temperature, P and EC (Kendall test: temperature, $\tau = 0.12$, $p$ value = 0.06; $SiO_2$, $\tau = −0.13$, $p$ value = 0.05; EC, $\tau = 0.11$, $p$ value = 0.09; and $P$, $\tau = −0.03$, $p$ value = 0.61).

### Mat thickness versus all the predictor variables

According to the RF model, the physical, chemical, temporal and biological variables (density and presence of *D. geminata* and *Cymbella* spp.) could explain up to 49% of the variance in the mat thickness (RMSE = 0.41, $R^2 = 0.49$), and the variables with the greatest power of discrimination were phosphorus and the *D. geminata* and *Cymbella* spp. density (Table 1). When we look at the data in detail (Fig. 7), the mat thickness peak occurred at lower $P$ values. We highlight that the mat thickness peak with higher $P$ values occurred exclusively on the C/D sampling points during the spring. Additionally, for *Cymbella* spp., the C/A mat thickness peak occurred during spring but overall had lower $P$ values.

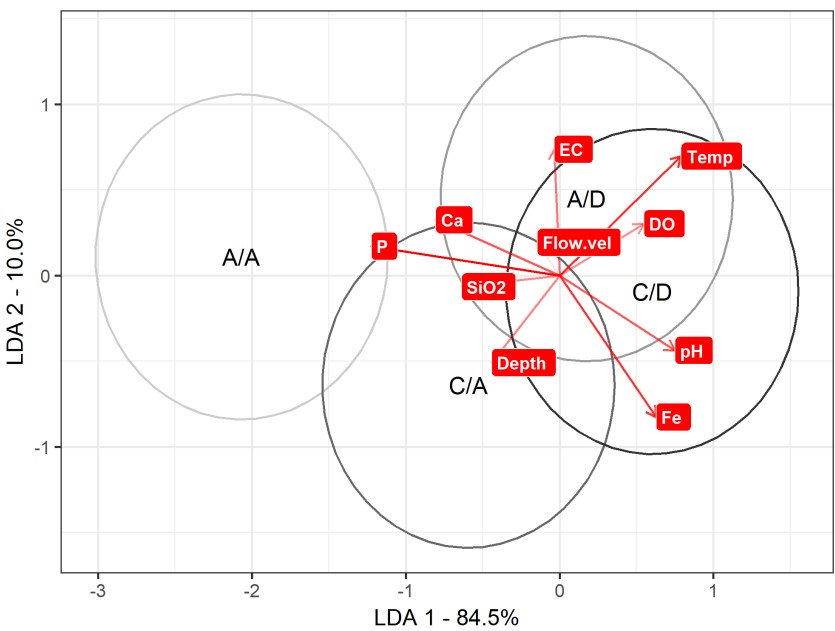

**Figure 5 LDA approach.** Result to LDA analysis. First axis explained 84.5% of variance, and second axis explained 10% of variance. Ellipse represent interval confidence estimated by Euclidian equation.

## DISCUSSION

### *D. geminata* and *Cymbella* spp. and their relationship with mat

The results show how the points had different mat thicknesses according to *D. geminata* and *Cymbella* spp. presence or absence. In the presence of both, the mat thickness was doubled. However, only with *Cymbella* spp. presence is it more likely that mats do not become visible (<0.5 mm). In contrast, only in the presence of *D. geminata* did the sampling points have a visible mat in most cases (median of 3 mm). This finding alone could suggest that *Cymbella* spp. does not have the ability to produce the massive growth that *D. geminata* does. This would imply that mat thickness duplication in the presence of both species does occur through an interaction between both species.

However, when evaluated on a temporal scale, the alternation of the mat thickness peak between the A/D and C/A sampling points in spring and autumn proves how each taxon separately can produce a thick mat and shows that mucilage production depends on both taxa, the physicochemical variables and time. Despite this, the C/D sampling points in both seasons had a stable mat thickness, which could suggest an interaction between the taxa or that the physicochemical variables at these points are advantageous for mucilage production. This last hypothesis agrees with different published papers (*Sundareshwar et al., 2011*; *Kilroy & Bothwell, 2012*; *Bothwell, Taylor & Kilroy, 2014*; *Montecino et al., 2016*).

Greater mat thickness at a lower cell density of *D. geminata* is consistent with the documented paradox between *D. geminata* and mucilage production, a process associated with low cell abundance (*Bothwell, Taylor & Kilroy, 2014*). For *Cymbella* spp., a greater mat thickness is present at intermediate densities, suggesting the existence of a cellular

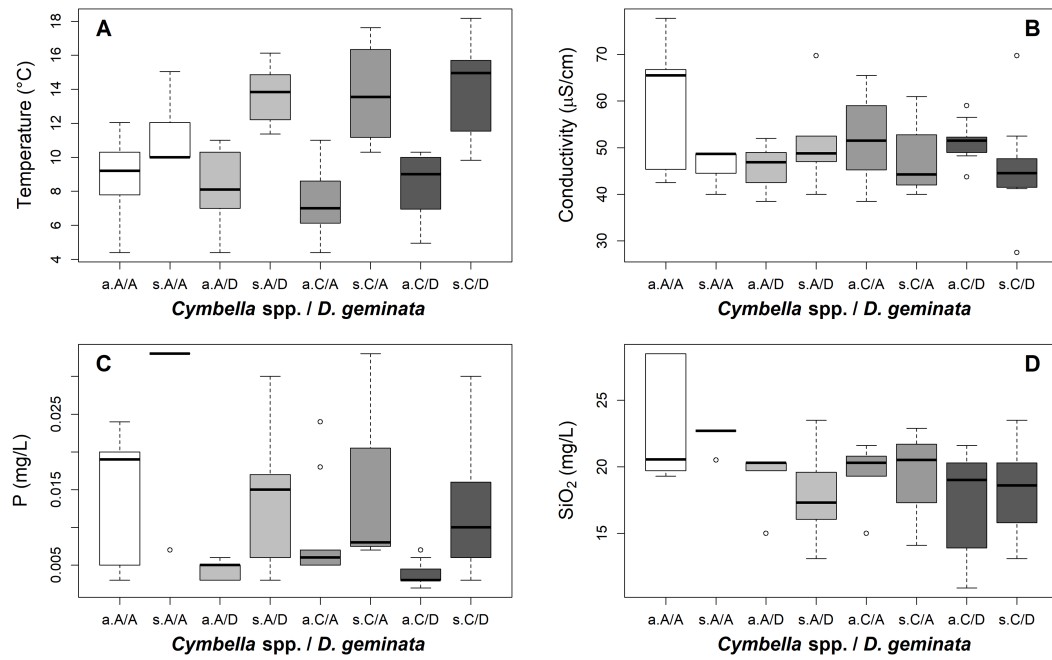

**Figure 6** **Boxplot of physical and chemical conditions per river reach sampled according to the presence/absence of *D. geminata* and *Cymbella* spp. and sampling season.** Samples were categorized as presence only (C/A, *Cymbella* spp. alone) or (A/D, *D. geminata* alone); the presence of both (C/D) and, the absence of both taxa (A/A). To sampling season, a, Autumn and s, Spring. (A) Water temperature. Sites in the presence of *D. geminata* differ from other sites during spring. (B) Water conductivity. Sites in the absence of both microalgae during autumn differ from the rest of the categories. (C) Total phosphorus. Sites in the absence of both microalgae differ from the rest of the categories considering both seasons. (D) Total silica. Sites in the absence of both microalgae during autumn differ from the rest of the categories. The rest of the chemical variables measured did not yield significant results.

optimal for the production of mucilage. These results suggest that *D. geminata* could develop mucilage in the early stages of colonization of a habitat or in situations of scarce nutrients, but not *Cymbella* spp., which would require a certain cell density to produce more mucilage. However, there are no studies investigating the production of mucilage associated with this genus.

For *D. geminata*, the results showed a temporal alternation between the cell density and mat thickness at the sampling points without *Cymbella* spp., which is in accordance with the literature (*Khan-Bureau et al., 2014*; *Bothwell, Taylor & Kilroy, 2014*) and demonstrates the temporal scale of this phenomenon. On the other hand, the results for *Cymbella* spp. showed that this taxon had an even higher cell density and mat thick in the presence of *D. geminata*, but without *D. geminata* presence the mat thickness peak occurred at a low cell density and during spring. This pattern could indicate how mucilage production in *Cymbella* spp. is more dependent on physicochemical conditions than its cell density. Currently, there is evidence that cell density alternates with *Cymbella* species (*Khan-Bureau et al., 2014*) but does not exist in relation to mucilage production, and thus, the hypothesis about these patterns in *Cymbella* spp. should be addressed in the future.

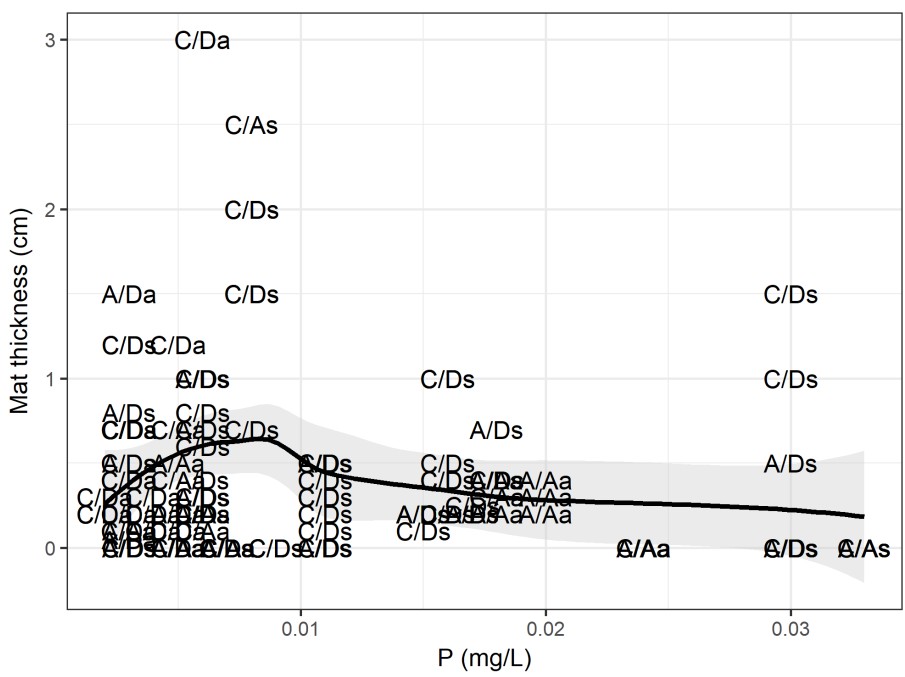

**Figure 7 Phosphorus and mat thickness.** Relationship between mat thickness and phosphorous. Each dot is a site categorized by presence only (C/A, *Cymbella* spp. alone) or (A/D, *D. geminata* alone), the presence of both (C/D) and, the absence of both taxa (A/A), and sampling season, a, Autumn and s, Spring. Black line represents GAM fit model, and gray area is interval confidence. The mostly sites with higher mat occurred during spring, with one exception during autumn.

Regarding the sampling point in the presence of *D. geminata* and *Cymbella* spp., we suggest that a high cell density of both taxon and a thicker mat during both seasons would be a consequence of their relationship through the mucilage: *D. geminata* with a lower cellular density would produce mucilage (increased mat thickness), particularly in autumn, while *Cymbella* spp. with an intermediate cellular density would produce mucilage in spring, alternating this work and maintaining a constant mat thickness. As we do not have a microscale count of the stalks, we do not have clarity on which species produces mucilage during spring or autumn in the C/D sampling points; however, a stable cell density suggests a population in good condition, probably due the mat thickness that generates suitable habitat for these taxa (*Ladrera, Gomà & Prat, 2018*).

Previous studies have documented the ability of *D. geminata* mats to generate habitat for small diatoms and other microorganisms (*Domozych, Toso & Snyder, 2010*; *Ladrera, Gomà & Prat, 2018*) by reducing the cutting effort through increased mat thickness (*Larned et al., 2011*; *Cullis, Crimaldi & McKnight, 2013*), which favors the colonization of cosmopolitan diatoms (*Kilroy, Larned & Biggs, 2009*) and is consistent with the pattern detected.

### The role of the physical and chemical variables on *Cymbella* spp. and *D. geminata*

Examining the physicochemical differences between the categories allow us to construct hypotheses about the niches of both taxa. First, in multivariate analysis, the A/D and C/D

sampling points were differentiated from the A/A and C/A sampling points, suggesting that the realized niche of *D. geminata* is near its fundamental niche, as it was recorded only on the sampling points with the physicochemical variables suitable for its presence. This result is striking considering that *D. geminata* is a novel invasive species in Chile (*Reid et al., 2012*; *Jaramillo et al., 2015*; *Montecino et al., 2016*). In contrast, the C/A sampling points not having significant differences from the other sampling points categories suggests that *Cymbella* spp. has not colonized all of its suitable sampling points, separating its fundamental niche from its realized niche.

This situation raises several questions about the dispersal abilities and historical context for these taxa. The more cited question is about the colonization process of *D. geminata* in Chile: did it arrive recently or was it always there? In the first instance, evidence supports that *D. geminata* apparently has been in Chile for longer than it seems (the first record was cited in *Asprey, Benson-Evans & Furet, 1964*). However, if we consider its dispersal and colonization skills, this pattern could also be the product rapid dispersion and difficult removal from the rocks that it colonizes. These questions are even more extensive in *Cymbella* spp., a genus that has just recently been investigated as mat producers both in Chile and in the world (*Suzawa, Seino & Mayama, 2011*; *Jaramillo et al., 2015*; *Khan-Bureau et al., 2016* among others).

While the temperature, P, silica and EC varied significantly between the sampling point categories, the temporal scale proves the plasticity of both taxa. For example, in spring, *Cymbella* spp. and *D. geminata* apparently could prefer high temperatures, though in autumn, the water in the A/A sampling points was equally as cold as the C/A, A/D and C/D sampling points. For EC and silica, the differences in values occurred only during autumn, when the A/A sampling points had the highest values compared to the other categories, but in spring, the silica and EC values were similar in all the categories. Only the P variable was consistent in select habitats for these taxa in both seasons, with ever lower values at points with *D. geminata* and *Cymbella* spp. presence. Thus, the results again show the relevance of phosphorus.

The richness of microalgae increased with the phosphorus and EC levels, which was contrary to the pattern recorded in *D. geminata* and *Cymbella* spp. This could be interpreted as a pattern that differentiates mucilage producers from other algae. This hypothesis can be supported by the river continuum concept (*Vannote et al., 1980*). Microalgae that are adapted to headwaters with low flow and watershed areas should need the ability to resist turbulent flow and low P levels, with mucilage production being a trait that could even be adaptive. On the other hand, microalgae adapted to river mouths would be adapted to low flow turbulence, low river substrates and higher P levels without needing mucilage production to attach to river substrates. However, much research is needed to confirm this hypothesis.

Finally, it should be noted that the patterns of silica, phosphorus and EC observed in the presence of *Cymbella* spp. and *D. geminata* has previously only been documented for *D. geminata* and phosphorus (*Bray et al., 2017*), which coincides with the results presented by *Spaulding & Elwell (2007)*, where a range of phosphorus levels in *D. geminata* has frequently been detected in the USA.

### The role of physical and chemical variables in mat thickness

We suggest that the relevance of phosphorus for mat thickness is strongly related to the presence of *D. geminata* and *Cymbella* spp. The principal antecedent for this hypothesis is the low P levels at the C/A, A/D and C/D sampling points. There is evidence of a relationship between P and *D. geminata* (*Kilroy & Bothwell, 2012*; *Bothwell, Taylor & Kilroy, 2014*). In this work, we add evidence that also relates phosphorus with mat thickness, but we now add *Cymbella* spp. as an additional taxon. The other variables that explained mat thickness in the random forest were the *Cymbella* spp. and *D. geminata* densities. This is relevant for highlighting the role of other taxa, such as *Cymbella* spp., on the debate about river mats.

The mat thickness recorded was much smaller than that of the mats in countries strongly impacted by *D. geminata*; one open question that this paper suggests is the role of *Cymbella* spp. in massive *D. geminata* blooms, since this would be relevant on mat volume and *Cymbella* spp. would compete with *D. geminata,* or it could possibly be releveled secondarily. In the second case, the importance of *Cymbella* spp. in this work could relate that physicochemical conditions do not allow *D. geminata* to make massive amounts of mucilage.

One interesting result is the peak mats at the A/D sampling points in autumn. Is it more favorable in Chile for *D. geminata* to produce mucilage in autumn? If the answer is yes, the tourists of these rivers would not visit during the worst mat levels months. Additionally, tourist visits would occur precisely during the highest flows in winter. Both conditions explain why *D. geminata* does not strongly impact Chilean rivers compared to other countries.

Because there are no limnological data linked to mat thickness in Chile, there is no information to compare or establish the ability of *Cymbella* spp. to produce mucilage, which highlights the novelty of the present study.

## CONCLUSION

The presence of *D. geminata* and *Cymbella* spp. increases mat thickness, and the temporal results suggest an alternation process of mucilage production by these taxa. While *D. geminata* showed a mat peak in autumn, *Cymbella* spp. showed a peak in spring with a constant thick mat at the C/D sampling points. The presence of both taxa and mucilage production are apparently strongly related to P and temperature levels, which is consistent with the international literature. The findings of this study suggest that it is necessary to evaluate the microalgae community together when we try to determine the relationship between mucosal development and species. Further research must continue to investigate the processes that determine the biological production of mucilage in rivers.

## ACKNOWLEDGEMENTS

Thank you to Magaly Olivares, Ursula Romero and Leonardo Núñez for their valuable comments and feedback.

### Funding

This work was supported by FIP 2013-25 project "Evaluation of Didymosphenia geminata (Didymo) in bodies of water in the south-central area," from the Subsecretaría de Pesca y Acuicultura through the Fondo de Investigación Pesquera, Chile. The funders had no role in study design, data collection and analysis, decision to publish, or preparation of the manuscript.

### Grant Disclosures

The following grant information was disclosed by the authors:
Fondo de Investigación Pesquera, Chile.

### Competing Interests

Dr. Matías Peredo-Parada is an employee of Plataforma de Investigación en Ecohidrología y Ecohidráulica limitada. Carilina Díaz is Chief of Amakaik Consultoría Ambiental.

### Author Contributions

- Daniel Zamorano conceived and designed the experiments, analyzed the data, prepared figures and/or tables, authored or reviewed drafts of the paper, approved the final draft.
- Matías Peredo-Parada analyzed the data, authored or reviewed drafts of the paper, approved the final draft.
- Diana J. Lillo conceived and designed the experiments, performed the experiments, authored or reviewed drafts of the paper, approved the final draft.
- Jorge Parodi authored or reviewed drafts of the paper, approved the final draft.
- Carolina A. Díaz conceived and designed the experiments, contributed reagents/materials/analysis tools, approved the final draft.

### Data Availability

All data collected are available in the Supplemental File.

### Supplemental Information

Supplemental information for this article can be found online at http://dx.doi.org/10.7717/peerj.6481#supplemental-information.

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
