# Peer review of "Mat thickness associated with Didymosphenia geminata and Cymbella spp. in the southern rivers of Chile"

_PeerJ, doi:10.7717/peerj.6481_

## Round 0.1 · original submission · Major Revisions

Major revisions are needed. Please address the reviewers comments in a revised submission. The language also need to be improved.

Reviewer 1 ·

Basic reporting

Please see attachment.

Experimental design

Please see attachment.

Validity of the findings

Please see attachment.

Additional comments

Please see attachment.

Annotated reviews are not available for download in order to protect the identity of reviewers who chose to remain anonymous.

Reviewer 2 ·

Basic reporting

• The English in the manuscript will need some editorial work to align single/plural, subject/verb, and editing some unclear statements. Replace weak verbs (i.e. is, are, were, be, etc) with active verbs to bring clarity and improve meaning in each section of the manuscript. Consider this an important edit. This will add more meaning to each sentence and will help you remove the repetition of ideas.
• Introduction: The most effective changes you could make to the introduction would be to bring in some literature to build a case for why we might care about mat thickness. Take the time to expand the information you allude to currently in the introduction. The strong case for your study that then emerges from this will draw your reader in more, and even broader the applicability of the study. Consider describing what we know about how mucilaginous mats for Didymo or other species changes flow, nutrient dynamics, insect emergence, and surface area available for epiphytes. Then, what might it mean if a mat is thick or thin.
• Is there a reason you might prediction interactions between diatom species that produce mucilage? How? Why? Anything unusual about Didymo here?
• Tables and Figures. In general these are Okay. Consider the following edits to improve clarity:
o Fig. 1. Captions should be stand alone. Be clear what the letters A, C, D represent (Even though it might seem intuitive). Does the black box in B represent the area zoomed in on in A? Be clear. In (C), doe
o Figs 2 and 3. – Italicize genus/species names in axis titles.
o Fig. 3. Remove repetition. For example … “ Mat thickness as a function of cell density of (A) Cymbella sp. and (B) Didymosphenia. Dropping one ‘l” in cell does not reduce much space. Instead use both l’s and add the ‘s’. For example “cells/mm3”
o Table 1 caption. – Line 3. “The percentage of the importance of a variable….”. Define your acronyms (for example, what does Vel fond” mean…velocity..??. Not clear what “not participate” means. Watch your subscripts for things like SiO2. Be consistent with use of capitals vs. small capitals in headings (i.e. MS residuals vs Var. Explained). Be clear that depth represents water column depth
o Table 1. Reduce clutter by putting units (like %) in the column header. This will also allow you to right align the numbers to allow for easier comparison between large and small numbers. Reduce the repeated column labels down to one column (this will also allow easier comparison).

Experimental design

2. Experimental Design - needs a few details
• L92 – You state that each sampling point was “analyzed” – please clarify what it was that you analyzed.
• L97 – Provide some details for how you subsampled each sample. Typically mucilaginous mats need to be blended. So how confident are you in your subsample. If samples were blended, did you conduct any microscope analysis of the samples prior to subsampling (i.e. to assess where the stalks were attached for each taxon etc.).
• 40x is a pretty low magnification for diatoms (but adequate to identify Cymbella and Didymo cells), so be clear that only the large Cymbella and Didymo cells were identified, but not the others.
• Since you later try to build the case for an interaction with Cymbella and Didymo, do you have any image evidence to show how Cymbella used Didymo stalks? Did you measure average stalk length in the different mate scenarios? This might make that argument more convincing.
• Provide a little more detail for nutrient and other elemental collection and analysis. I.e. were the samples taken near the mats? Above the mats? Below the water surface? Kept cold? Frozen? Analyzed within X number of hours, etc.

Validity of the findings

Results
• To improve clarity throughout the results, remove repeated phrases and replace “weak” verbs with active verbs. For example….” Of the 36 samples, detection of cells by microscopic counts occurred for D. geminata in 13 river reaches in the spring (___%) and 17 in the autumn (____%), and for Cymbella in 14 river reaches in the spring (___%) and 18 in the autumn (___%). However, of the ## of sampling points…..”…etc.
• Line 138. Report the range of mat thickness for mat >1cm.
• Line 140: Clarify what the 36 represents. (The 36 that contained Didymo?? or both? or…. Mats , of a certain thickness…or of the 36 river reach samples??)
• Results should be in **past tense.
• It also helps to follow the same “structure” within a paragraph. For example in L144 to 150, it looks like you want to talk about mats with Didymo alone, Cymbella alone, or both together. Thus, L147 states….. “…with the absence of both genera….”, so the next sentence, L149 should read “However, each taxon alone (C/A or A/D) did not differ from mats with both taxa absent (A/A) or both present (C/D)”. Present in this ‘parallel’ fashion throughout the results. Acronyms can be helpful, but the message is more effective if you refer to the mats with a taxon alone or present together and put the acronyms afterwards if it helps.
• Lines 144 to 150. Be clear on mat content. For example. “….in the presence of Cymbella sp. alone…”.
• Line 152, Add units after the 0.
• Line 155: “….in the presence of both genera (C/D)
• Line 153. Put this summative statement first…then follow with the details.
• Line 162 – Move the interpretation of the results to the discussion.
Discussion
• The discussion needs more literature comparison. This lack of discussion with other literature weakens the strength of this section and reduces the breadth of readers the manuscript might reach. Consider this a one of your top edits. This section needs some attention.
• Go through the discussion to align grammar and subject verb agreement. For example. Line 196 –remove the “s” from present or changed to “presented”. It will also help to replace weak verbs (such as is, was, were, etc) with active verbs throughout. (i.e. L197. – just state “In the presence of both taxa, mat thickness doubled. However mats typically occurred as thin (< 0. 5 mm), less visible biofilms with Cymbella sp alone, in contrast with the visible, thicker (median 3 mm) mats more commonly present with D. geminata alone.”
• You make several bold statements about what you think your data state about Cymbella, but the evidence is not that strong. If you want to make an argument for a few scenarios that might stem for your data, then back the statements up with supporting literature. For example.
o ***L200 Your data does not assess that your Cymbella sp. does not have the ability to produce massive growths. Perhaps it would be helpful to shed some light on your observations here by comparing with the literature. Cymbella can produce mats in other systems, so it certainly is quite capable.
o L207. Given that Cymbella was present in thin mats, this implies that they are present in early colonization stages, so it is not clear why you think that Cymbella might not be a colonizer. With respect to a certain cell density required for mat formation – is there some evidence of this in other taxa? A mechanism that might be at work? Etc.
o You attempt to build a case for the role of mucilage in increasing surface area for Cymbella. Any evidence to support this? The taxa that typically are reported to colonize Didymo stalks are often small (as you mentioned), and very visible under the microscope. Perhaps include an image plate showing stalk structure with Didymo alone, Cymbella alone or both.

Additional comments

The ideas in the manuscript will be of interest to those interested in Didymo and other stalk forming diatoms. Some care editing around language and time spent really setting the stage for the role of mucilage will help
Misc. General
References: Do a careful check for italics of genus and species names (for example Furey et al. ref.; Ladrera et al. ref.; Kilroy et al. ref.).
Do a spell check for D. geminata vs. D. geminate. If your sentence begins with “D. geminata”, write out Didymosphenia in full (i.e. Line 39 in abstract). Check also for italics (i.e. L 73)
Correct typo in title (…associated..)
Replace the word “community” with algal biofilm (or some other word) if you are only referencing algae. Community suggests more than one trophic level. (I.e. Methods – L92.

·

Basic reporting

The reporting is professional with some minor grammatical issues, otherwise this paper is well thought out and written. Please correct. See the list.

Minor grammatical issues:
1. Line 47 D. geminate is misspelled should be geminata
2. Line 53 mats producers should mat producers
3. Line 61 Cymbella should be italic.
4. Line 65 Cymbella should be italic.
5. Line 86 – on two occasions
6. Line 126 D. geminate is misspelled should be geminata
7. Line 127 one and two way ANOVAs were…..
8. Line 195 Spell out Didymosphenia
9. Line 199-200 correct grammar.
10. Line 201 due should be do
11. Line 213 doubles should be double
12. Lines 237-238 I would not use dashes after the numbers.
13. Line 239 thick should be thickness check grammar.
14. Check all taxa to ensure the names are italic.

Question: Were there any dams or impediments upstream from the blooms? Were the rivers regulated in any way?

Experimental design

The experimental design is appropriate.
1. The methods can be replicated.
2. The research question is relevant and interesting.
3. The methods are described sufficiently.
4. Very nice maps and details.

Validity of the findings

Data is sufficient and related to the authors questions. Question - What Cymbella taxa did you find in the different location samples?

Data is robust and sound.

Additional comments

Important issues:
1. Recommend a couple of images of Didymosphenia geminata and Cymbella sp. cells. LM and only if possible SEM if available.
2. Recommend 1 or 2 photos of the mucilaginous stalks laid out or being held to see the length and 1 photo of the stalks on the rocks.
3. Why Cymbella sp.? Can’t you ID the different species? Don’t make sp. italic if you continue to use sp.
4. Recommend that you use sp. only when you are unsure of the species (Khan-Bureau at al. 2014).
5. I would recommend to write Cymbella taxa instead of sp. unless you do not know which Cymbella species are present.
6. Boulders – you may want to say how large those boulders are. This is why I would have a photo.
7. It is important to distinguish the Cymbella taxa. You could also mention whether the Cymbella taxa that you studied are native and which taxa are not.
8. Were all the Cymbella taxa that were mentioned found in all of the samples? If not distinguish which Cymbella taxa were in the different samples.

This paper is interesting and appears that the experimental design and methodology is appropriate.
I find your conclusions regarding water temperature and nutrients interesting. Typically Cymbella janischii or Cymbella taxa blooms at different times under different conditions than D. geminata. (Khan-Bureau et al. 2014, 2016) However with that said habitat are diverse and continually changing (i.e. climate change and adaptation etc.). I do agree that further work needs to be done to determine if your conclusions of mucilage production are representative of river systems with the presence of both D. geminata and Cymbella taxa.

Please correct your use of sp. in your manuscript. Sp. is used when a species cannot be verified although can be identified to genus level. Some scientists use cf. when comparing like species. I recommend that you distinguish which Cymbella taxa were found in each sample and recommend writing Cymbella taxa instead of Cymbella sp. I recommend that you provide a photo of the mucilaginous stalks of Didymosphenia geminata with Cymbella taxa on the same rock, substrate or location. I recommend that you have LM and if possible SEM images of D. geminata and Cymbella taxa cells. These items should be corrected prior to acceptance.

---

## Round 0.2 · Major Revisions

A further major revision is needed. Please pay particular attention towards improving the language so that the science can be better understood.

Reviewer 4 ·

Basic reporting

While the questions addressed in this manuscript are interesting, it suffers from a poor translation into English. With the Introduction, Methods, and Results I have tried to make as many grammatical comments as I could. But with the Discussion I had a very hard time following the author’s points and I did not want to inadvertently change the intended meaning of the text. Rather, you will need a friendly reader to go over that section to clarify your points. Until then, I am not able to fully judge the merits of this paper. Given the very short turnaround time given to review this manuscript, this is the best I could do.
The Intro seems well-referenced with many of the latest related studies. Although at one point the author mentions that no studies have been conducted on the stalk production for the genus Cymbella- While I cannot provide additional references, I believe that such papers exist and would encourage the authors to try another search to find them.
The overall structure of the paper seems fine.
The figures and tables seem good in general. I have made some comments below on a few specific points.
The raw data provided seem in order and appear to support at least the main points (as I understand them) of the manuscript.

Experimental design

This work is original and within the scope of the journal. The research question is interesting, well defined, relevant and meaningful. The methodology seems sufficiently rigorous. And the method (in most cases) are provided with enough details to replicate the findings.

Validity of the findings

As a reviewer, I had trouble (due to poor application of English grammar) assessing some of these points. But in general, the authors took an interesting approach to an understudied natural phenomenon that could lead to management implications.

Additional comments

-It is clear that a lot of work went into the development of this manuscript. I am hopeful that the English language can be improved so that some of your conclusions can be more clearly understood by a wider audience. This manuscript would benefit by the authors going over it again with an English editor. In many sections, I am not always sure of the authors intent and so I am reluctant to suggest edits that might change the author’s meaning.
-The way you cite references in the text is not standardized and often has many mistakes. If you are citing a paper with more than three authors use et al. after the first author!
-Please explain in the method why these specific rivers were chosen for the study. How typical are these rivers to others in the region? Do other rivers seem to have more or less Didymo associated with them? Why or why not? Were the same sites samples during spring and fall?
-please provide more details on how the algae samples were actually collected. As written, I do not understand the procedure at all.
-Please provide more statistical information. Include the type of permutations used along with the software (including version) used.
-In most cases, when giving the mean value of a parameter you have measured, its range should also be reported.
-you mention total richness of taxa in the results but we need more details on how this was calculated in the methods. What was the minimum number of algal cells counted before it was determined a taxa was absent? Were other taxa identified and counted to determine the “total richness”? If so, how many were counted? If not, do not call this variable “total richness”.
Specific comments:
Line 18- change to “have shown chemical…”
Line 26- change to “… sites with Cymbella spp. Presence showed a spring maximum.”
Line 38-change to “…mats can…”
Line 41- change to (Bray et al., 2017) but also check this reference since as reported it is not in the literature cited.
Line 45- change to Larned et al, 2011)
Line 47-48- I cannot tell what you are trying to say with this sentence.
Line 49- suggest changing to “the presence of nitrogen-fixing cyanobacteria (Novis et al, 2016) and levels of dissolved reactive phosphorus (Bothwell et al., 2014 and Bray et al., 2017) would favor D. geminata blooms…”
Line 55- “This problem” –not sure what you mean. “who have begun to warn by others possible bloom maker microalgae”-I have no idea what this means. Please clarify.
Line 61- change to “been reported”…
Line 91-94-“one algal sample of 1ml of mat was taken”-Did you mean to report an area (such as 1cm2) here instead of a volume? I don’t understand this. Please elaborate on these methods! Within a reach, how was the specific site(s) selected? This is especially important since it seems that in some cases just one sample was taken for an entire river reach. Please clarify the information for the rest of this passage and give a final concentration for the Lugol’s solution in the samples.
Line 105-change to “the laboratory within 48hours.”
Line 109-My understanding is that there is more than one way to conduct these permutations. If that is the case please provide more info. Also be clear about what two dependent factors you are using.
Line 110- Which “non-parametric Tukey test” was used. Please be specific.
Line 115- REML? -please reference any software mentioned- along with the version used.
Line 116-change to “different plots”…
Line 120- delete “in the first place” and change to “To relate physical…”
Line 124-125- What was the name of the test used?
Line 125-129- As written, these lines are very hard to follow. The grammar is especially poor here.
Line 134- change to …”evaluated with R2”. But also need more information here. What R2s were considered not significant? Does this need to be justified?
Lines 135-142, Good! But much of this information should be included as soon as a specific test is first mentioned. Again, include package version and version of R used.
Line 142- change to “macroscopic photos…)
Line 152-157. Please consider reporting this data as a table rather than in text.
Lines 159-162-Please report if any of these differences were significant.
Lines 162-167. Was this the twoway ANOVA? If so, were there any significant interactions?
Line 169- change to “inverse relationship”
Line 177-change to “density was always relatively high”…
Line 182- I suggest NOT reporting f-values.
Line 188- change to “detected significant differences…”
Line 215- “total richness of taxa” How was this calculated? I don’t think that this was mentioned in the methods.
Figure 1-good figure! It may be helpful to have the rivers on the map marked with darker lines. Its hard to see them.

---

## Round 0.3 · accepted · Accept

I carefully checked the revised version, and I am satisfied with the revision.

#